# Factors Influencing COVID-19 Vaccine Uptake among Nepali People in the UK: A Qualitative Study

**DOI:** 10.3390/vaccines10050780

**Published:** 2022-05-14

**Authors:** Padam Simkhada, Pasang Tamang, Laxmi Timilsina, Bibha Simkhada, Paul Bissell, Edwin van Teijlingen, Sunil Kumar Sah, Sharada Prasad Wasti

**Affiliations:** 1School of Human and Health Sciences, University of Huddersfield, Huddersfield HD1 3DH, UK; p.p.simkhada@hud.ac.uk (P.S.); pasang.tamang@hud.ac.uk (P.T.); laxmi.timalsina@hud.ac.uk (L.T.); b.d.simkhada@hud.ac.uk (B.S.); 2University Management Team, University of Chester, Chester CH1 4BJ, UK; p.bissell@chester.ac.uk; 3Faculty of Health and Social Sciences, Bournemouth University, Bournemouth BH1 8GP, UK; evteijlingen@bournemouth.ac.uk; 4Mid Yorkshire Hospitals NHS Trust, Leeds Teaching Hospitals, Leeds WF1 4DG, UK; sunil.sah@nhs.net

**Keywords:** COVID-19, vaccine, vaccine uptake, Nepali community, qualitative research

## Abstract

Vaccination saves lives and can be an effective strategy for preventing the spread of the COVID-19, but negative attitudes towards vaccines lead to vaccine hesitancy. This study aimed to explore the factors influencing the uptake of the COVID-19 vaccine in the Nepali community in the United Kingdom (UK). This qualitative study included in-depth interviews with 20 people from Nepal living in the UK. Interviews were conducted by a native-Nepali speaker and all interviews were audio-recorded, transcribed, and translated into English before being analysed thematically. Our study found that attitudes towards COVID-19 are generally positive. Nine overlapping themes around barriers to COVID-19 vaccination were identified: (a) rumours and mis/disinformation; (b) prefer home remedies and yoga; (c) religion restriction; (d) concern towards vaccine eligibility; (e) difficulty with online vaccine booking system; (f) doubts of vaccine effectiveness after changing the second dose timeline; (g) lack of confidence in the vaccine; (h) past bad experience with the influenza vaccine; and (i) worried about side-effects. Understanding barriers to the uptake of the COVID-19 vaccine can help in the design of better targeted interventions. Public health messages including favourable policy should be tailored to address those barriers and make this vaccination programme more viable and acceptable to the ethnic minority communities in the UK.

## 1. Introduction

COVID-19 spread rapidly around the globe in 2020 and was declared a pandemic by the World Health Organization (WHO) on 11 March 2020. The global cumulative data show a disproportionate burden of infections and deaths in selected countries [1]. Vaccination is one of the most successful public health interventions and a cornerstone in the prevention and control of vaccine-preventable diseases, including COVID-19, to reduce morbidity and mortality [2,3]. By December 2021, a total of 30 COVID-19 vaccines were approved globally, ranging from a single dose to a maximum of four doses [4]. However, the availability of vaccines does not guarantee their use. Worldwide, vaccine hesitancy is one of the biggest threats to public health, as seen for a variety of vaccines such as the polio vaccine in, for example, Nigeria and Pakistan [5,6], the measles vaccine in Europe and North America [7,8], and the influenza vaccine in 2009 [9]. Vaccine hesitancy is influenced by several factors including the behavioural decision to accept, delay or reject it, perhaps because of not trusting a vaccine or a provider, or not perceiving a need for or value of the vaccine [10]. The consequences of hesitation towards vaccinations are potentially harmful to both individuals and their community, but studies show that community-level vaccine coverage requires 80% and above to protect the community, depending on vaccine efficacy and the duration of protection [3]. However, vaccine hesitancy or refusal is one of the most important health threats [3,11,12]. 

COVID-19 vaccination programmes have started globally following the approval and administration of multiple vaccines, and in the United Kingdom (UK) this started in December 2020 [13]. One year later, by the end of December 2021, 8.99 billion vaccine doses had been administered globally, resulting in 57.3% of the world’s population having received at least one dose of a COVID-19 vaccine, of whom 48% are fully vaccinated. Moreover, more than two-thirds (69.1%) of the UK population were fully vaccinated by December 2021 [4,14,15]. Despite this progress in vaccine uptake, several studies raised concerns about vaccine hesitancy and reluctance to their use [3,16,17,18]. Some evidence suggests that the willingness to vaccinate against COVID-19 may change over the period, and the hesitancy and lower acceptance levels challenge the success of vaccination programmes [19]. The global COVID-19 vaccine acceptance rate was 71.5%, which ranged from almost 90% (in China) to less than 55% (in Russia) [20]. Likewise, a recent systematic review of 28 countries revealed that only 60% of respondents were intended to be vaccinated and 20% would refuse vaccination [21]. Similarly, a recent Chinese study revealed that around half (48%) of respondents would delay the vaccination until the vaccine’s safety was confirmed [17]. However, around 36% of people in the UK [22], 51% in the United States [3], 59.9% in Turkey [23] and 22% in Japan, are either uncertain about getting vaccinated or unlikely to be vaccinated against COVID-19

Ethnic minorities in the UK are less likely to intend to seek vaccination, e.g., Black (28%) or Pakistani/Bangladeshi (57.7%) groups were more hesitant compared to Whites (16%) [24]. However, it is not clear what the reasons are behind such a lower willingness and intention to seek COVID-19 vaccination in several ethnic minority groups [18]. There is a paucity of literature on factors influencing the uptake of the COVID-19 vaccine, and the few reasons reported include side-effects, misinformation regarding vaccine efficacy, being worried about unknown future effects of the vaccine, not trusting the vaccine and antivaccine sentiments [25,26], but it is not clear what factors impede the uptake of COVID-19 vaccines in the Nepali community living in the UK. It is estimated that over 80,000 people of Nepali origin are living in the UK, with retired British Gurkha soldiers and their families being the largest sub-group in this population. A recent study shows that there is a lack of understanding in this community of the UK’s health and social care system [27]. 

Similarly, a recent systematic review of UK studies focusing on ethnic minority groups and COVID-19 vaccine hesitancy, found no studies on Nepali people residing in the UK [28]. On the other hand, in the early days, the situation around COVID-19, as well as vaccinations, were very unclear, and the perceptions of different ethnic minority communities seem to differ from each other. At the same time, the UK Government has been struggling to improve the uptake of COVID-19 vaccines in several sub-groups in the population. To make vaccination programmes successful, it is important to understand the willingness to accept vaccines, as well as to develop effective policies and health promotion messages to maximise the uptake of the vaccine. Therefore, this study aimed to explore the factors influencing the uptake of COVID-19 vaccines in Nepali people living in the UK.

## 2. Methods and Materials

### 2.1. Study Design and Sites

The study was qualitative using one-to-one interviews. Data were collected between January and March 2021. Qualitative research is the ideal method to gain in-depth and detailed narratives about personal topics [29]. This study was conducted in London, Kent, Liverpool, and Aberdeen (Scotland) as these four locations have a relatively high Nepali population.

### 2.2. Participant Selection and Data Collection Methods

A total of 20 participants from three different stakeholders were interviewed, including ten interviewees from the Nepali community; seven health workers (doctors, nurses, pharmacists, care assistants); and three representatives of organisations supporting the Nepali community. Study participants were purposively selected [30] based on health status, gender, locality (living in England or Scotland), and age. A list of potential participants was prepared with the help of researchers, health care workers, and community stakeholders. Potential participants were contracted through email and phone. After they agreed to be interviewed, researchers conducted in-depth interviews virtually (using phone, Zoom, Microsoft Teams). Interviews were conducted with the aid of interview guides to elicit information about the participants’ views on COVID-19 vaccination [30]. All interviews were conducted and recorded in Nepali and interviews lasted 20 to 30 min. 

Interview guide:What do you think about the COVID-19 vaccine? If offered today, will you be willing to take the COVID-19 vaccine?If you have already been vaccinated, what motivated you to take the vaccine?In your opinion, why do you think that people come for the COVID-19 vaccination? Why don’t they come for the COVID-19 vaccination?

### 2.3. Data Analysis

Audio recordings were translated into English by two Nepali researchers. Data were analysed iteratively and inductively using a thematic approach [30]. The verbatims presented in this paper best represented the key themes. The factors influencing vaccine hesitancy are explained based on the epidemiological triad and are a complex interaction of environmental- (external) agent- (COVID-19 vaccine) and host-specific (vaccine recipient/people) factors (Figure 1) [26]. During the analysis, the themes were revised multiple times until the research team reached a consensus. 

### 2.4. Ethical Consideration

The study protocol was approved by the Institutional Review Board of the University of Huddersfield, UK. Consent was obtained verbally, and participants’ anonymity and confidentiality were assured. No personal identifiers were recorded in the electronic dataset to maintain anonymity and the quotes are also identified by health care workers, ordinary Nepali people in the community, and stakeholders, alone, with a serial number designated by ‘P’. 

## 3. Results

This paper starts with participants’ backgrounds followed by the motivation to present for vaccination and barriers to the uptake of the COVID-19 vaccine. Most (70%) were male, with an average age of 46 years, and half of all participants were vaccinated against COVID-19 (Table 1).

### 3.1. Motivation for the COVID-19 Vaccination

Most participants who had been vaccinated reported as their motivation: (a) personal safety; (b) safety of other people; and (c) trust in science/reducing the COVID-19 pandemic.

(a)Personal safety

Personal safety often was the key reason for getting a vaccination. Participants spoke of people dying with COVID-19, as well as being excited to get vaccinated themselves. 


*I’ve seen the family being wiped out with the infection. I’ve seen the devastation. This virus works in a very strange way, we are slowly starting to understand … I don’t want them to be hospitalised and wanted to get vaccinated as soon as possible … Thankfully, I got my first vaccination. So, hopefully, in the next few weeks, I’ll get my second one as well.*
(P 8, Health Worker)

A health worker stated that the COVID-19 vaccine was regarded as rigorously tested and compared it with childhood immunisations for preventable diseases. 


*I would strongly advise everybody to get the vaccine. These have been rigorously tested and now millions of people have been vaccinated. So, we have crossed seven million in terms of vaccination in the UK. So, nobody’s been admitted or had any serious side effects. The vaccine is safe; that was the motivation to go for [the] jab. *
(P 12, Health Worker)

(b)To save other people

Participants also saw being vaccinated as protecting others, not just their own health. Participants gave examples making life-and-death comparisons and explained that the risk of dying from COVID-19 was higher than the side-effects of vaccination.


*I took vaccines as a part of infection prevention measures. I have to keep myself safe so that I can keep my family safe as my wife is with me now. The main motivation was for infection prevention. *
(P 3, Health Worker)


*People also know that there might be side-effects, but the risk of dying is more dangerous than the side-effects, and [we should] promote the vaccine using local community leaders. *
(P 11, Community Leader)

(c)Trusted on science and evidence to control the pandemic

Participants with a health background gave their trust in science as the major reason for COVID-19 vaccination. They said, for example, that the vaccine underwent a robust approval process and they were confident in its safety. 


*The COVID-19 vaccine has been made after many studies and the government has declared it safe, so I do not see any reason to be worried. The vaccine will protect us as individuals as well as reduce infection. *
(P 11, Community Leader)

Participants gave several examples where COVID-19 could not be prevented/controlled, but they confidently said that COVID-19 can only be controlled through vaccination programmes, especially where many preventive measures had failed to control it.


*I believe in science and the only way we can get out of this is to vaccinate. We have waited and waited and waited, the herd immunity [herd immunity occurs when a large portion of a community becomes immune to a disease, making the spread of disease from person to person unlikely] never happened …. social distancing and lockdown did not work much. So, the only way out is now getting everybody vaccinated. *
(P 6, Health Worker)

A health worker linked childhood and influenza (flu) vaccination to the COVID-19 vaccine:


*I was better informed because I know about these vaccines. I take my flu vaccine every year. I’ve done that for 20 years now. I had probably had all childhood vaccinations. I know how rigorously the [vaccine] production [is, and] the safety and their approval processes are. So, I didn’t have any doubt whatsoever [and] that fostered me to take jab. *
(P 2, Community Leader)

### 3.2. Barriers to the Uptake of the COVID-19 Vaccine

Findings identified a range of factors that negatively influenced the uptake of the COVID-19 vaccine among the Nepali community in the UK. The result is presented under three key themes: (1) environmental/eternal factors; (2) agent/vaccine-specific factors; and (3) personal factors. These three themes comprised a total of nine sub-themes (Figure 1). 

(1)Environmental/External Factors

Participants spoke about several environmental- or external-related factors that impeded the use of the COVID-19 vaccine.

#### 3.2.1. Rumours and Dis/Misinformation Regarding the Vaccine

Easily accessible misinformation through social media is one of the biggest threats to public health programmes. Participants repeatedly reported that rumours or dis/misinformation about the COVID-19 vaccines make people reluctant to go for vaccines. Most interviewees equivocally blamed online news portals and social media (Facebook and YouTube) for spreading rumours against the COVID-19 vaccine. Participants spoke of misinformation (inadvertent) and disinformation (advertent) as a threat, especially during COVID-19 outbreaks, and that the misleading information tends to spread faster than accurate information through social media. 


*I know that in our media, our social media, the South Asian media, negativity sells quicker, and they have been there pointing out the negative side of the vaccine more than the positive things of it. *
(P 6, Health Worker)

Participants repeatedly highlighted that the internet and social media such as Facebook and WhatsApp are used to share anecdotal experiences and promote anti-vaccine campaigns. 


*I think some people are just being bombarded with misinformation over social media. *
(P 2, Community Leader)

Social media were regarded as fuelling negative messages which makes people hesitant to get vaccinated. 


*There [is] lots of fake news over WhatsApp, Facebook and in everything. So, I think this social media is making another trouble. Yes, my wife asked the same question when she went for the vaccine. *
(P 14, Ordinary Nepali Person in the Community)


*I was looking at Facebook earlier today as well, a lot of my friends who are talking about side-effects, you know, what sort of side-effects did you get and they’ve started a poll as well saying, you know, like, with lots of symptoms, saying minor, major or serious symptoms but it’s very, very rare. *
(P 8, Health Worker)

Rumours on the side-effects of the vaccine, and reports of it being derived from animal products such as pork, confused people and such rumours added to their reluctancy:


*Some people might not be getting their answer like me, some White colleagues at my workplace fear … [a] long-term side-effect that might happen, some Muslim colleagues at the beginning said they don’t want [it] because the vaccine is made out of pork. *
(P 9, Health Worker)

Participants not only raised concerns about side-effects but said people did not fully trust the current vaccine trial as it was conducted in such a short time, implying trials were conducted in a hurry:


*There is multiple information in social media where it says that the trial has not been completed for the vaccines and the long-term effect of the vaccine is still uncertain. So, there are a group of people who say that due to these reasons, the vaccine shouldn’t be taken. *
(P 3, Health Worker)

#### 3.2.2. Home Remedies and Yoga

Nepal has many traditional medicines and people opt to follow traditions from Nepal, using some of these in the UK. All participants, including the health practitioners, described using home remedies (*turmeric, ginger, Ajawin* [carom seeds] *water, basil leaves, pepper tea, steam inhalation, yoga, massage with mustard oil)* as preventive measures against COVID-19. Drinking enough hot water, siting in the fresh air, doing regular exercise and yoga were reported by many to help boost the immune system. A further few claimed that using different home remedies and yoga and exercise also helps to avert COVID-19 transmission. 


*Most people are using home remedies such as turmeric, ginger water, pepper tea, drinking plenty of water, massage at night with mustard oil. Open windows every day to let the fresh air in, sleep on time and do exercise regularly. *
(P 16, Ordinary Nepali Person in the Community)


*I take raw turmeric, Ajawin, cumin seeds with warm water in a big glass and then go to work. I also take steam every day. Again, after coming from work, I take [a] shower and then follow the same home remedies. Sometimes I take basil leaves, ginger and many other home remedies. I use turmeric every day twice a day with Ajawin, cumin seed, [and] ginger with warm water. *
(P 9, Health Worker)

Many participants suggested that such home remedies are not limited to people of Nepali origin, but they gave examples of British people who had also suggested taking home remedies instead of vaccines.


*My neighbour was saying that his medical doctor, who is also British, suggested to take home remedies rather than vaccination. So, I was shocked to hear that there is a medical practitioner who doesn’t believe in vaccination. *
(P 3, Health Worker)

Participants further suggested the regular use of yoga to boost the immune system:


*I do meditation every morning, after doing that my body feels so relaxed. I do [it] every morning and maintain the balance of my body. That helps me to avoid stress when I go to work. I feel relaxed and this helps to keep anxiety and stress levels very low. When we do such yoga and meditation, then there is a development of positive hormones in our body. *
(P 9, Health Worker)

#### 3.2.3. Religious Restrictions

A few participants spoke of religious restrictions. They believe that vaccine is made from a serum, which was deemed to be a barrier to Muslims taking it.


*Well, this country [UK] is full of cultures. Some of them religiously can’t take a serum and vaccinations, so, people have a diverse opinion about the vaccine. Probably that’s why they don’t want to take this. *
(P 13, Health Worker)


*Very few people, they don’t believe [that] vaccines work for COVID-19 which may be the reason from their culture and religion. *
(P 2, Community Leader)

#### 3.2.4. Concern about Vaccine Eligibility

Participants raised concerns regarding the COVID-19 vaccine eligibility group at the time, as the vaccine was not offered to all age groups, pregnant women or breastfeeding mothers. One breastfeeding mother gave it as a reason for not getting vaccinated:


*I’m breastfeeding at the moment and should not be vaccinated when you are breastfeeding. So, I’m hoping to get one when my baby will have solid food and I would not be breastfeeding. I might get it after six-months. *
(P 13, Health Worker)

#### 3.2.5. Difficulty with the Online Vaccine Booking System

Participants gave one practical difficulty experienced by ethnic minority groups in their attempts to get the COVID-19 vaccine, namely the online appointment system which is not practical for the elderly and people with poor English and/or low literacy rates. This difficulty is linked with the large proportion of British Gurkha families who are living in the UK. 


*To take a vaccine is not easy in the UK where we need to take an appointment which is not easy for all our [Nepali] community. You know that a large proportion of the Nepali community in the UK are Gurkha families [retired British army] which is not easy to make an appointment and vaccinate due to the system. *
(P 3, Health Worker)

Participants raised the concern on appointment difficulties due to language barriers and that the current COVID-19 restriction of one-person per health care consultation meant that people who don’t speak English could not be accompanied by a relative. As one participant explained:


*I do not understand the guideline given by the Government as they are in the English language. I do not know much English, and my wife does not know any English at all. We have a big problem when we go to General Practitioners. They provide health information in many local languages, but we do not have it in Nepali. I can speak some English but when my wife has appointments, they ask to come one person only so it’s difficult for her as she does not understand English at all. *
(P 4, Ordinary Nepali Person in the Community)

Participants further put the concerns about the delay in the COVID-19 vaccine appointment process, which they found impracticable:


*While booking an appointment we cannot understand the English properly so we cannot ask for an interpreter too. Before COVID-19, we could go to the clinic and book in person, which was easy, but phone booking takes a long wait and [it is] very difficult to understand English too. It took me 1 h 45 min last week to book an appointment for a vaccine. This is not practical [for] everyone. *
(P 4, Ordinary Nepali Person in the Community)

(1)Agent/vaccine-specific factors

Participants raised concerns around vaccine-specific factors that impede the use of the COVID-19 vaccine.

#### 3.2.6. Doubt of Vaccine Effectiveness after Changing the Second Dose Timeline

Few participants expressed doubts regarding the changed timing of the second dose. The interval between the first and the second dose was originally three weeks, as advised by scientists for its effectiveness. However, the UK government lengthened the time to the second dose time to 12 weeks, which confused people and raised issues about the vaccine. 


*The most important thing … is the conflicting information that the British government has been providing and irritating people. For example, the leaflet that we are given during the vaccination says that both the doses of vaccine should be taken within 21 days otherwise it won’t be effective. However, the government made a policy where people will get the second dose in 12 weeks with the intention that the first dose has to cover a maximum number of people. A medical scientist has been saying that the maximum gap is six weeks. After that, there aren’t any benefits of the vaccine from first to second doses. But the government ignored the scientific evidence and brought a rule of [a] 12-week gap between the doses. So, when such conflicting information comes to the media then people get very much confused about the decision. *
(P 3, Health Worker)

Perceptions of vaccine effectiveness were also grounded in misconceptions about how long COVID-19 vaccines prevent the transmissions and how effective they were. This confusion led to a wait-and-see mentally, whereby people waited for more information about the currently offered vaccine.


*I have a question about [the] vaccine which I have not got answered yet, how long will this vaccine work? Whether it will work for one year, 10 years or forever? I didn’t get an answer to this question. Now, there is another thing, when this vaccine was developed it was COVID-19, but now there are different new variants. So, I don’t know whether it will work for this or not. *
(P 9, Health Worker)

(1)Host/Person-Specific Factors

Range of individual or person-specific factors were reported by participants that impede the use of the vaccine.

#### 3.2.7. Lack Confidence in the Vaccine

Participants knew about the vaccine as an effective disease control measure and there was no doubt about the reality of the COVID-19 pandemic. Despite knowing all the benefits, participants stated that some of their work colleagues were not interested in taking the vaccine because they do not feel the currently offered COVID-19 vaccine is effective. A female community participant stated that:


*One of my staff didn’t want…the vaccination and I asked the reason and she said this is her choice and she doesn’t feel safe. *
(P 14, Ordinary Nepali Person in the Community)

One male community participant also did not know anybody personally who had received both doses of the vaccine who was therefore protected: 


*I heard from someone or read somewhere that this vaccination does not protect us against COVID infection, rather it only prevents us from getting worse from infection.….I have not found anyone who has [been] vaccinated and is well protected from COVID-19. *
(P 16, Ordinary Nepali Person in the Community)

Participants further questioned whether the currently offered vaccine works against the regularly changing strains of COVID-19 in the UK. This also makes people hesitant to go for the vaccine.


*It is also unsure whether [the vaccine] will work [for] the new strain or not. This is also one of the reasons for people not believing [in] the vaccines. *
(P 3, Health Worker)

Participant further narrated that the currently offered vaccine was completed in a very short period (see also the section entitled ‘rumours and dis/misinformation regarding vaccine) and that it might have side-effects, which people want to observe first and then decide whether or not to go for the vaccine.


*I have not taken a jab yet. I am not saying that I don’t want to, but I want to see the side-effects and how it will go. It has been developed so fast and is used so quickly. I am just wondering that it might affect everyone at once when it’s used. At present, I have no fear and I am managing well. I will take the vaccine a little later. I have three questions that are not clear to me about this vaccine. So, I am delaying [it]. *
(P 9, Health Worker)

#### 3.2.8. Past Bad Experience with the Influenza Vaccine

Some participants spoke about their negative experiences with influenza vaccines in the past, which stopped them from booking COVID-19 vaccine appointments. This poor experience with an influenza vaccine also meant ignoring appointments for influenza vaccination, despite this being a priority for National Health Service (NHS) health workers. The participant further stated that:
*Ram [name changed] in his late 40′s has been working in [the] NHS for more than 10 years as a health support staff [member]. Since the start of the COVID-19 pandemic, he has worked in contact with COVID-19-positive cases admitted to the hospital for treatment. But Ram is reluctant to take the COVID-19 vaccine. Despite being the priority group for the vaccine programme as a frontline worker, he says that he doesn’t want to take the vaccine and has been “escaping from his vaccination”. His supervisor has been continuously doing follow-ups for the vaccine. However, Ram has been postponing it every time, saying that he will take it later. Ram has had the flu vaccine in the past, and he suffered from severe side-effects of the flu vaccine. He was ill for more than 2 weeks after taking the flu vaccine. He had [a] fever for about two to three weeks and some people told him that he developed [a] fever due to the flu vaccine. Since then, Ram completely left out taking the flu vaccine. Ram fears that the COVID-19 vaccine will do the same to him and he doesn’t want to go for a jab because of the fear of side effects. He heard, as well as [he] read somewhere, that the currently offered vaccination does not protect against COVID-19 infection, rather it only prevents from getting worse from infection only. It saves [you] from getting worse, [and does] not protect [you] from getting the infection. Steve [name changed] also told Ram that he took the first dose of the vaccine and felt sick, and he has been asking Ram for suggestions on how to skip the second dose, which doubled Ram’s fear of vaccine side-effect. Ram believes in following natural processes like sleeping on time, eating on time, eating healthy food, doing regular exercise, drinking plenty of water and avoiding alcohol. Ram believes that this will help him against COVID-19 attacks. Ram will only take the vaccine if there are rules which say that people who do not get vaccinated will have travel restrictions, or if everybody in the UK is vaccinated except him. When Ram sees everyone is safe after getting the vaccine, then he will feel confident that nothing happens and will gain trust towards the vaccine.*(P 16, Health Worker)

#### 3.2.9. Being Worried about Side-Effects

Many participants raised their fear of side-effects. Few participants who were vaccinated experienced side-effects such as tingling in the hand, having a sore arm, getting chills, feeling very cold, and having a fever within the two weeks after vaccination. They were not sure about going for their second dose. 


*I did get some side-effects with the second dose of the vaccine for about two days. I was getting chills, very cold and unwell but I have fully recovered. Nothing serious happened. *
(P 12, Health Worker)

A member of the medical staff further stated that:


*My wife and I … our arm was very, very sore the next day, which was fine. *
(P 8, Health Staff)

In addition, a few participants who were in priority categories for COVID-19 vaccination hesitated and had not taken the vaccine yet. 


*I have heard mixed feelings about the side-effects of the vaccine. Some people say they have mild pain and flu-like symptoms. Some people say it’s only niggling in the hand, not much really. Some people have headaches and [are] feverish. *
(P 13, Health Worker)

A middle-aged woman working as a care home manager said that she had witnessed side-effects of COVID vaccine in her care home patients, as well as colleagues:


*I have heard and seen immediate side-effects of the vaccine. I have seen-side effects on my other colleagues, [and] older people at the care homes. I wanted to see but I don’t know anything about the long-term side-effects. Lots of other vaccines are coming and other vaccines worked so far, [more] than the current one. *
(P 9, Health Worker)

Others were afraid of side-effects of the current COVID-19 vaccine, as well as participants also linking the side-effects, they faced during the influenza vaccine (see above). Few participants further narrated that vaccination had, sooner or later, resulted in side-effects therefore they are not wanting to go for a COVID-19 vaccination.


*The side-effects that I experienced were fever after 2–3 weeks. Some people said that I developed a fever due to the flu vaccine. So, I completely left out taking the flu vaccine. I felt it was better without a vaccine. *
(P 16, Community Leader)

A community leader further added that:


*Very few people may worry about [the] jab because they might have a severe allergic reaction to the vaccine. *
(P 2, Community Leader)

## 4. Discussion

This study is the first of its kind to explore factors both positively and negatively affecting the uptake of a COVID-19 vaccine in this UK ethnic minority group. Among a diverse group of 20 study participants, perceptions, reasons for the vaccination, and barriers to opting for the vaccine (when available) were explored. Understanding all factors is important for appropriate messaging from the vaccination team and policymakers, as well as to help design targeted interventions for this community. The discussion section starts with the motivations to take a COVID-19 vaccine, followed by the barriers.

### 4.1. Motivations to Take a COVID-19 Vaccine

Participants who had been vaccinated, or were waiting to be vaccinated, were driven first and foremost by notions of personal safety, and to prevent the transmission of the COVID-19 virus to other people. A recent study conducted in China also shows that the perceived severity of COVID-19 was positively associated with the motivation to get vaccinated [31]. People took a COVID-19 vaccine to save themselves and others, and to reduce its transmission, and hence prevent COVID-19 [32]. It shows that people who understood the benefits of vaccines for an individual and society had decided to receive a vaccine. A meta-analysis study also shows that fear and perceived efficacy are important determinants of behaviour intention and behaviour change in public health interventions [33]. Recent UK studies revealed that people with positive beliefs and attitudes towards vaccines, and who want to protect others from catching COVID-19 and want to get back to normal life, tend to be vaccinated when a vaccine becomes available [22,24,25]. 

Many of our participants trusted vaccine development and its evidence base, and global efforts towards controlling the COVID-19 pandemic. Most participants were aware of these efforts and they gave various childhood vaccines and their lifesaving characteristics as examples. Therefore, a large number of participants who were waiting to get their vaccination commented on the vaccine’s importance and the robust production process. A recent study into global trends and barriers to the uptake of vaccines conducted in 149 countries, between 2015 and 2019, also shows that people who understand the importance of vaccines were more likely to take up vaccination [12]. Moreover, participants who had a higher belief that the vaccine would be safe were more likely to say they intended to take the COVID-19 vaccine [24]. Therefore, public health messages should be tailored to outlining the benefits and importance of vaccines from a trustworthy source of information, to this minority ethnic community.

### 4.2. Barriers to Uptake of the COVID-19 Vaccine

Barriers to taking a COVID-19 vaccine emerged in three broad range of themes: (1) environmental factors (rumours and mis/disinformation, preferring home remedies and yoga, religious restrictions, concerns regarding vaccine eligibility and difficulty with the online appointment system); (2) vaccine-specific factors (doubts about vaccine effectiveness/changed timing of the second dose); and (3) personal factors (lack of confidence in the vaccine, past bad experience of the influenza vaccine and being worried about side-effects).

(1)Environmental factors

Almost all participants were willing to get their vaccination but only three participants stated they were not interested, for several reasons. A study conducted between November to December 2020, revealed that only 18% of UK participants were unlikely or very unlikely to go for vaccination, whereas a higher proportion of ethnic minorities such as Black (71.8%), Pakistani and Bangladeshi (42.3%), other background (26.4%) participants were “unlikely or very unlikely” to go for vaccination compared to White British (15.6%) participants [24]. 

Easily accessible misinformation (inadvertent) and disinformation (advertent) is one of the biggest threats with social media, as participants repeatedly reported that, due to overwhelming rumours about vaccine efficacy and its usefulness, as well as its side-effects, make people more reluctant to take the vaccine. It was evident that misinformation (inadvertent) and disinformation (advertent) are not a novel threat to public health, especially during disease outbreaks [34]. People are desperate for information to assess their risks of getting the disease, its severity, and possible preventive and curative measures but the evidence is equivocal that the misleading information has the tendency to spread faster than accurate information through various outlets [35]. Evidence on misleading information on popular social media platforms (e.g., YouTube, Facebook, online newspapers) during the COVID-19, Ebola and Zika outbreaks, reported that at least one-quarter of popular content (in terms of shares, likes, visits) was misleading [36,37]. Similar findings also noted in an earlier study, conducted elsewhere, that the world has been affected by a massive “infodemic” where an over-abundance of information, which involves the excessive dissemination of false information associated with COVID-19, that makes people more confused about deciding on vaccination [38,39,40]. Health professional and vaccine advocates may curb the spread of misinformation on social media to obviate further deepening the false opinion through using the various social media platforms in terminating the spread of misinformation. 

Our findings revealed that older adults had difficulty with online appointment systems for the COVID-19 vaccine because of low literacy and English language difficulty. It is reported that around 80% of the Nepalese people residing in the UK are ex-Gurkha soldiers and their families. Similar communication issues were also reported in a study conducted within Nepali communities, reporting that a lack of understanding of the NHS system impeded access to health and social care services [27]. It is also evident that making the vaccination appointment process easy and offering the public choices of locations, increases uptake and meets the goals of COVID-19 vaccination campaigns [41]. Therefore, the COVID-19 vaccine appointment system should develop a pragmatic way for older adults and non-native speakers to fulfil policies related to the uptake of services. 

(2)Vaccine-specific factors

Our findings revealed that some people prefer home remedies and yoga as COVID-19 preventive measures, rather than receiving the vaccination. Furthermore, participants worried about the COVID-19 vaccine and its efficacy due to the negative messages and misinformation. Vaccine safety concerns were reported in Hong Kong, where 78.4% were worried about safety and 63.2% had doubts about the effectiveness of the vaccine [42]. From the beginning, there were conspiracy theories surrounding the COVID-19 pandemic and vaccines. Rumours existed about the virus being man-made, and that the vaccine could lead to infertility and limit the growth of the human population [43,44]. Similar findings were also revealed in the UK and USA, that misinformation is strongly associated with declines in vaccination intent [40]. Most respondents of a Portuguese study also reported that they would like to wait some time or a long time before taking the COVID-19 vaccine [45]. Public willingness to accept a vaccine is therefore not static but it is highly responsive to the current information about vaccine effectiveness, as well as the perceived risk of contracting the disease. A study in China showed that over 90% of the respondents stated that they would accept the COVID-19 vaccine when it became available, but almost 50% of these people wanted to delay their vaccination until it was confirmed to be safe [17]. Likewise, a study conducted in Italy also revealed that the available vaccines may cause severe health consequences and people also worried about the vaccine effectiveness [46]. Therefore, it is crucial to build confidence regarding COVID-19 vaccine, as its perceived safety and efficacy, related to vaccine intention and hesitancy, should be constantly monitored and evaluated in order to change strategies as deemed necessary. In addition, targeted information should be released through trusted sources of information. The findings show that the currently offered COVID-19 vaccine has a high level of side-effects, and this can give other diseases rather than protect against the COVID-19 virus. On the other hand, participants who had a bad experience with the influenza vaccine in the past also perceived the severity of its side-effects and they are not willing to take a vaccine. Perception of the severity of the disease, fear of side-effects, and concern about efficacy was reported as the possible factors affecting the uptake of COVID-19 vaccination in previous studies [24,42]. 

Low confidence in the COVID-19 vaccine and inconsistent and contradictory information provided by the Government makes people hesitate to take the COVID-19 vaccine. Findings revealed that the timeline between the first and second dose of the vaccine was within three weeks for its effectiveness, but the government has changed the second dose time up to 12 weeks and made people more confused and cautious about receiving the vaccine. Participants raised concerns and doubts about the information the public can trust, science and evidence or political information. The contradictory messages regarding the first and second dose and the safety and efficacy of the COVID-19 vaccine raised big concerns for the study participants. A similar finding occurred in a longitudinal study about vaccine hesitancy in the UK, where 42.7% of respondents reported that they were worried about the unknown future effects of the vaccine [24]. Moreover, a recent study conducted with US healthcare workers also reported that around two-thirds (63.6%) of respondents were worried that the current vaccination may be ineffective against new strains [47]. Similarly, a previous study also recommends building confidence in the COVID-19 vaccine as its perceived safety and efficacy was strongly associated with people’s intention to take the vaccine [45]. Therefore, the Governments and health authorities should increase the trust of the community, where people have false beliefs and fears that make them hesitant to go for the vaccination. The findings also further raised concerns on vaccine effectiveness, which was also grounded in misconceptions about how long the currently offered COVID-19 vaccines work to prevent COVID-19, and their ability to control the daily changing strains of COVID in the UK. Although evidence shows that COVID-19 vaccines are effective in reducing hospitalisation and death from COVID-19, and prevent its transmission, it is also recommended that non-pharmaceutical interventions such as social distancing and wearing face masks shall need to remain in place for the vaccinated population, for the overall effectiveness of the vaccine [48].

(3)Personal factors

Our findings revealed that almost all participants felt more concern about the fear of the side-effects of the COVID-19 vaccines, rather than the benefits. It is evident that most vaccines have side-effects, but COVID-19 vaccines are very new, and the real side-effects are unknown. This fear is understandable and prevalent, but it is, therefore, extremely important to ensure everyone receives high quality information about COVID-19 vaccines, in terms of not only their efficacy but also their side-effects. A similar fear regarding the side-effects of the vaccines was also reported in many studies conducted elsewhere, other than COVID-19 vaccines, with diverse communities [45,49,50,51]. Findings revealed that, participants have very mixed feelings about the COVID-19 vaccine and think that the vaccine has been produced and endorsed very rashly, which makes people plan to wait and see how it goes with others before they decide whether to take the vaccine. Similar findings were also reported in a study conducted in the USA [32]. Lack of knowledge and information on immunisation may impede the vaccination program’s success [49,51]. Although, it is a known fact that the efficacy of any approved COVID-19 vaccine is in the range of 70–80%, and no vaccine has 100% efficacy, the rush of vaccine production with higher risks of safety issues may jeopardise community trust and lead to hesitancy in the uptake of services [52]. Therefore, knowing detailed information regarding the potential side-effects and coping mechanisms may foster a greater uptake of the services. This study’s data collection was carried out during the early months of 2021, when the vaccination campaign was in its very early stages. Therefore, it is likely that many people, especially so in ethnic minority contexts (which typically are less trusting of public authority), could be indifferent or preoccupied with regard to the message of vaccine safety and side-effects.

### 4.3. Strengths and Limitations

To the best of our knowledge, this is the first qualitative study on the Nepali community residing in the UK about the reasons why a COVID-19 vaccine would be accepted or refused. This study was timely as COVID-19 vaccine development and its hesitancy among minority groups in the population was widely discussed in the media. The UK government, nationally, had rolled out a vaccination programme and our findings provide evidence on one minority group which needs targeted interventions to foster a community vaccination programme in the future.

This study does have some limitations. The study participants were purposively selected; something that may have been exacerbated by the use of a virtual platform, thus, the findings should be generalised cautiously. As the researchers facilitated the recruitment of participants through their personal and professional contacts, this may have restricted the range of opinions reported. Hence, selectivity bias exists of people who were included. These study findings might be influenced because of the early period of the study conducted, where the vaccination campaign was just initiated, and could overestimate actual concerns from the ethnic minority community that could have emerged from a study conducted later on. Finally, most participants were young, therefore the views of the elderly Nepali people in the UK may not be fully represented by our research.

## 5. Conclusions

With the growing Nepali community in the UK, uptake of the COVID-19 vaccine is necessary and probably should be made mandatory for the success of the vaccination program. A range of socio-cultural and language barriers, a complicated appointment process, the fear of side-effects, and concerns regarding effectiveness and efficacy issues were reported, that directly impact the uptake of vaccination services among the Nepali community in the UK. Education and favourable policy should be made available to this community to support a successful vaccination programme. The vaccine appointment process should be made easier or individual general practice centres could make appointments that may foster COVID-19 vaccination. Although vaccine hesitancy, particularly in minority ethnic communities, is higher than in the general UK population, perhaps our study in the Nepali community can offer insights into considerations of other minority groups, especially from the South Asian continent. Therefore, this appears to be a potentially fruitful avenue for future studies with a larger sample size, to validate our results and provide better insights into the underlying reasons for vaccine acceptance and its contributing factors. 

## Figures and Tables

**Figure 1 vaccines-10-00780-f001:**
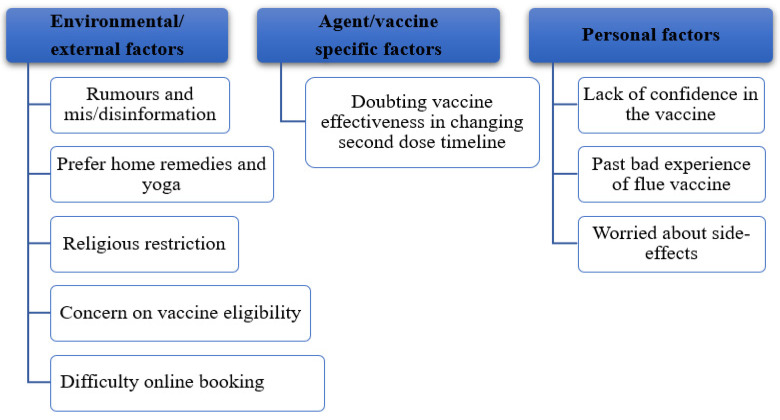
Key themes around barriers to take the COVID-19 vaccine.

**Table 1 vaccines-10-00780-t001:** Participants Characteristics.

Variables	Number	%
**Gender**		
Male	14	70.0
Female	6	30.0
Age (Median age)	46 years	
Age ranges in years	34–72	
**Participants’ affiliations**		
Retired /Gurkha’s family)	10	50.0
Health worker (doctor, nurse, pharmacist, care workers)	7	35.0
Community leaders	3	15.0
**Vaccination status**		
Vaccinated	10	50.0
Nonvaccinated	10	50.0

## Data Availability

The datasets generated during and/or analysed during the current study are not publicly available due to the sensitive nature of data but are available from the corresponding author on reasonable request.

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
