# Peer review of "Factors Influencing COVID-19 Vaccine Uptake among Nepali People in the UK: A Qualitative Study"

_vaccines, 2022, doi:10.3390/vaccines10050780_

Round 1

Reviewer 1 Report

I congratulate the authors for carrying out a very thoughtful and effective revision of the paper. In this new form, the paper is written considerably batter and is also methodologically acceptable, as the authors have revised their claims according to the remarks and have made a much more substantial use of the ethnographic material to support their claims. Also the discussion section is considerably improved and now provides interesting insights.

My only remaining suggestion is to better justify in the introduction why it is important to study the Nepali community and to introduce the issue a bit less out of the blue -- a reference to the Nepali community comes up in the middle of a more general sentence without a proper preparation. But well done. 

Author Response

Reviewer Dear,

 Thank you very much for the useful feedback on our paper (Vaccines-1618157).  We have addressed all minor issues mentioned in the review and we have indicated below the changes in below response sheet.  As a corresponding author, I would like to confirm that the manuscript has been read and agreed for the submission by all the named authors.

Reviewer one 

1. My only remaining suggestion is to better justify in the introduction why it is important to study the Nepali community and to introduce the issue a bit less out of the blue -- a reference to the Nepali community comes up in the middle of a more general sentence without a proper preparation

Thank you for your valuable feedback. We have added few sentences in the respective section.

On the behalf of corresponding author,

Dr. Sharada P Wasti

Reviewer 2 Report

Thank you for a well-written study that adds to the body of evidence. 

Line 40 - Nigeria is not the only country with vaccine hesitancy to Polio vaccine. Please update 

Line 49 .... mass vaccination for COVID-19 (specify COVID)

Table 1 - not sure why numbers are missing from some of the rows. Also cannot add IQR in the % column 

From lines 130

Quotes should be identified as such with quotation marks and perhaps italics font 

Line 368-369 - amend grammar slightly to be clearer. Should it be e.g. For one participant...

Line 447-449 important to include the dates of the survey carried out which was close to first wave and a different period to when your study was carried out: 24th November to 1st December 2020

Noted re retired soldiers and their family grouped together - were any of these family members born in the UK or educated in the UK for example? There might be differences in the barriers they experience? Also were the females members of the community, community groups or healthcare workers. I think this might also be important to include in the results briefly

Provide interview schedule and questions used as supplementary - others should be able to replicate the study from methods provided  

In conclusion - "the Nepali community seems to differ from other minority groups".

I am not sure this is totally true  please amend. The reasons/barriers for hesitancy are very similar to other groups. However, for some groups, the data available are heightened and they have been studied wider and broader. 

Is below a combination of explanation and quotes from the participant? Important to make clear please as a few areas needed edits which I have started below. If they are quotes, please use quotation marks in relevant places 

Ram (name changed) in his late 40s; has been working in NHS for more than 10 years as a health support staff.  Since After the start of the COVID-19 pandemic, he has worked in contact with COVID-19 positive cases
admitted to the hospital for treatment. But Ram is reluctant to take on taking the COVID-19 vaccine.
Despite being the priority group for the vaccine programme as a frontline worker, he says that he
doesn’t want to take the vaccine and has been "escaping from his vaccination". His supervisor has
been continuously doing follow-ups for the vaccine. However, Ram has been postponing it every
time, saying that he will take it later. Ram has had the flu vaccine in the past, and he suffered
from severe side-effects of the flu vaccine. He was ill for more than 2 weeks after taking the flu vaccine. He had fever for about two to three weeks and some people told him that he developed fever due to the flu vaccine. Since then Ram completely left out taking the flu vaccine. Ram fears
that the COVID-19 vaccine will do the same to him and he doesn’t want to go for a jab because of despite the fear of side effects. He heard, as well as read somewhere that the currently offered vaccination
does not protect against COVID-19 infection rather it only prevents from getting worst from infection only. It saves from getting worst not protecting from getting the infection. Steve (name changed) also told Ram that he took the first dose of the vaccine and felt sick, and he has been
asking Ram for suggestions on how to skip the second dose which doubled Ram’s fear of vaccine
side-effect. Ram believes in following natural processes like sleeping on time, eating on time,
eating healthy food, doing regular exercise, and drink plenty of water and avoiding alcohol. Ram
believes that this will help him against vaccine attacks. Ram will only take the vaccine if there are
rules which say that people who do not get vaccinated will have travel restrictions or if everybody
in the UK is vaccinated except him. When Ram sees everyone is safe after getting the vaccine,
then he will feel confident that nothing happens and will gain trust towards the vaccine (P 16-
Health Worker). 

Author Response

Dear Reviewer,

Thank you very much for the useful feedback on our paper (Vaccines-1618157).  We have addressed all minor issues mentioned in the review and we have indicated below the changes in below response sheet.  As a corresponding author, I would like to confirm that the manuscript has been read and agreed for the submission by all the named authors.

I have online uploaded my manuscript and look forward to hearing from you.

Thank you.

Yours faithfully,

Dr. Sharada Prasad Wasti

...........................................................................................

Response Sheet

Reviewer’s comments

Authors’ reply

Reviewer 2

Line 40 - Nigeria is not the only country with vaccine hesitancy to Polio vaccine. Please update 

Thank you so much for your suggestions, we have added Pakistan too.

Line 49 .... mass vaccination for COVID-19 (specify COVID)

Thank you so much for your suggestions, we have specified it.

Table 1 - not sure why numbers are missing from some of the rows. Also cannot add IQR in the % column 

Thank you so much for your suggestions we have removed IQR from the table.

From lines 130 Quotes should be identified as such with quotation marks and perhaps italics font 

Thank you so much for your suggestions, we have corrected it.

Line 368-369 - amend grammar slightly to be clearer. Should it be e.g. For one participant...

Thank you so much for your suggestions, we have revised it.

Line 447-449 important to include the dates of the survey carried out which was close to first wave and a different period to when your study was carried out: 24th November to 1st December 2020

Thank you so much we have added the study period.

Noted re retired soldiers and their family grouped together - were any of these family members born in the UK or educated in the UK for example? There might be differences in the barriers they experience? Also were the females members of the community, community groups or healthcare workers. I think this might also be important to include in the results briefly.

All the study participants were born in Nepal, but they have been working in different parts of the UK during the study period. We have not found exceptional different views as per the gender or participant’s professional background.

Provide interview schedule and questions used as supplementary - others should be able to replicate the study from methods provided . 

Thank you very much for your suggestions which has now added in the method section.

In conclusion - "the Nepali community seems to differ from other minority groups". I am not sure this is totally true please amend. The reasons/barriers for hesitancy are very similar to other groups. However, for some groups, the data available are heightened and they have been studied wider and broader. 

Thank you for your suggestions, we have edited accordingly.  

Is below a combination of explanation and quotes from the participant? Important to make clear please as a few areas needed edits which I have started below. If they are quotes, please use quotation marks in relevant places 

 Ram (name changed) in his late 40s; has been working in NHS for more than 10 years as a health support staff.  Since After the start of the COVID-19 pandemic, he has worked in contact with COVID-19 positive cases admitted to the hospital for treatment. But Ram is reluctant to take on taking the COVID-19 vaccine.
Despite being the priority group for the vaccine programme as a frontline worker, he says that he doesn’t want to take the vaccine and has been "escaping from his vaccination". His supervisor has
been continuously doing follow-ups for the vaccine. However, Ram has been postponing it every time, saying that he will take it later. Ram has had the flu vaccine in the past, and he suffered from severe side-effects of the flu vaccine. He was ill for more than 2 weeks after taking the flu vaccine. He had fever for about two to three weeks and some people told him that he developed fever due to the flu vaccine. Since then Ram completely left out taking the flu vaccine. Ram fears that the COVID-19 vaccine will do the same to him and he doesn’t want to go for a jab because of despite the fear of side effects. He heard, as well as read somewhere that the currently offered vaccination does not protect against COVID-19 infection rather it only prevents from getting worst from infection only. It saves from getting worst not protecting from getting the infection. Steve (name changed) also told Ram that he took the first dose of the vaccine and felt sick, and he has been asking Ram for suggestions on how to skip the second dose which doubled Ram’s fear of vaccine side-effect. Ram believes in following natural processes like sleeping on time, eating on time, eating healthy food, doing regular exercise, and drink plenty of water and avoiding alcohol. Ram believes that this will help him against vaccine attacks. Ram will only take the vaccine if there are
rules which say that people who do not get vaccinated will have travel restrictions or if everybody in the UK is vaccinated except him. When Ram sees everyone is safe after getting the vaccine,
then he will feel confident that nothing happens and will gain trust towards the vaccine (P 16-Health Worker). 

Thank you for your suggestions and few editing. We have thoroughly reviewed and edited the copy of the manuscript.
